STTM: an efficient approach to estimating news impact on stock movement direction

Riabykh Aleksei 1
Surzhko Denis 1
Konovalikhin Maxim 1
Koltcov Sergei skoltsov@hse.ru 2
1 Department of Data Analysis and Modeling, VTB Bank , Moscow , Russia
2 Laboratory for Social and Cognitive Informatics, National Research University Higher School of Economics , St. Petersburg , Russia
Maguitman Ana
Electronic publication date: 2022 Dec 16
Publication date: 2022
Volume: 8
Electronic Location ID: e1156
Received 2022 Jul 22; Accepted 2022 Oct 24
Copyright: ©2022 Riabykh et al.
Copyright year: 2022
Copyright holder: Riabykh et al.
License: This is an open access article distributed under the terms of the Creative Commons Attribution License, which permits unrestricted use, distribution, reproduction and adaptation in any medium and for any purpose provided that it is properly attributed. For attribution, the original author(s), title, publication source (PeerJ Computer Science) and either DOI or URL of the article must be cited.
License URL: https://creativecommons.org/licenses/by/4.0/

Keywords: Stock movement, Topic modeling, Time series, Stock markets, Sharpe ratio, Granger causality test

Funding: Basic Research Program at the National Research University Higher School of Economics in 2022 This work was supported by the Basic Research Program at the National Research University Higher School of Economics in 2022. The funders had no role in study design, data collection and analysis, decision to publish, or preparation of the manuscript.

==============================
Open text data, such as financial news, are thought to be able to affect or to describe stock market behavior, however, there are no widely accepted algorithms for extracting the relationship between stock quotes time series and fast-growing textual representation of economic information. The field remains challenging and understudied. In particular, topic modeling as a powerful tool for interpretable dimensionality reduction has been hardly ever used for such tasks. We present a topic modeling framework for assessing the relationship between financial news stream and stock prices in order to maximize trader’s gain. To do so, we use a dataset of economic news sections of three Russian national media sources (Kommersant, Vedomosti, and RIA Novosti) containing 197,678 economic articles. They are used to predict 39 time series of the most liquid Russian stocks collected over eight years, from 2013 to 2021. Our approach shows the ability to detect significant return-predictive signals and outperforms 26 existing models in terms of Sharpe ratio and annual return of simple long strategy. In particular, it shows a significant Granger causal relationship for more than 70% of portfolio stocks. Furthermore, the approach produces highly interpretable results, requires no domain-specific dictionaries, and, unlike most existing industrial solutions, can be calibrated for individual time series. This makes it directly usable for trading strategies and analytical tasks. Finally, since topic modeling shows its efficiency for most European languages, our approach is expected to be transferrable to European stock markets as well.

Introduction

Effective market hypothesis (EMH) (Fama, 1970; Fama, 1991) argues that all publicly available information is immediately and fully reflected in stock market prices. Consequently, neither historical data nor the forecasts based on them are seen as usable for the development of efficient investment strategies. However, many approaches for stock market movement prediction that were developed since EMH had been proposed (Fabozzi & Fabozzi, 2020) have demonstrated certain levels of efficiency. At the same time, as the task remains challenging due to the high volatility of stock quotes, new approaches are still needed. Overall, two main groups of approaches—technical and fundamental—are usually singled out by researchers, both nowadays employing machine learning methods (Dixon, Halperin & Bilokon, 2020). In technical analysis, the analyst uses past trends in the share prices to predict their performance in future, without inferring the causes of the observed trends. Fundamental analysis is based on the assumption that the market price of an asset tends to its intrinsic value, but always deviates from it with the asset thus being either overvalued or undervalued. By inferring the intrinsic value to which the market is expected to correct, this approach aims to predict stock price behavior. For this, various external data are often used, including information disclosed by companies, such as revenues, earnings or profit margin, and independent analytics.

One of the promising types of external information is unstructured textual data, notably financial news. Coupled with automated machine learning techniques, it allows investors to solve predictive and descriptive tasks, saving time and labor costs for finding important information in a large amount texts. Such data is found able to generate interpretable and significant information signals that help investors to minimize investment risks.

Shallow feature based methods of text processing play a special role in predicting the direction of different types of financial movement, such as stock or commodity prices, with unstructured text data, such as news or user-generated content. Most often, these methods do not require markup (unlike approaches based on sentiment analysis) and do not need updating their parsing algorithms (unlike event extraction methods). The general procedure of building such algorithms begins with preprocessing of the source texts, then passes to constructing vector representations, or embeddings of these texts (e.g., TF-IDF, BoW, Doc2Vec, DL-based embedding) and finally incorporates these embeddings in machine learning (ML) techniques to predict stock trends. The main disadvantage of such approaches is low interpretability of vector text representations as predictors. Meanwhile, topics generated by probabilistic topic models are easily interpreted by humans based on the lists of most probable words, but are mostly missing from the relevant literature reviews (Usmani & Shamsi, 2021; Jurczenko, 2020; Shah Dev & Zulkernine, 2019). Other dimensionality reduction methods that do find their way into financial movement prediction domain are mostly based on hard clustering approaches, e.g., K-Means (Babu, Geethanjali & Satyanarayana, 2011). This is suboptimal for classification of texts that usually belong to more than one topical cluster. Additionally, such clusters are difficult to interpret as they are delivered unlabelled. As topic modeling co-clusters both texts and words by topics, top words can be used as natural cluster labels, while simple clustering yields nothing except lists of items grouped into the unlabelled clusters. Although K-Means-based approaches can be ideologically adapted to fuzzy logic and to the logic of simultaneous co-clustering of items and their features, we are unaware of such applications in the sphere of stock market prediction.

In this article, we propose a new method for predicting stock price movement direction based on topic modeling. Our algorithm is highly interpretable, requires no fixed markup or pre-existing sentiment dictionaries, and at the same time remains an end-to-end solution within the paradigm of machine learning techniques for stock prediction using numerical and textual data. Our approach achieves high predictive power in the weekly price trend prediction task, where stocks of the largest Russian companies are considered as time series (spanning eight years between 2013 and 2021), and economic news of the three largest Russian-language news agencies are used as textual data. We use the Granger causality test to evaluate statistical significance of the obtained predictions. In addition, we consider a simple trading strategy and evaluate the success of a portfolio calibrated on the obtained predictions through Sharpe ratio and annual return. In doing so, we consider portfolios derived from predictions of various ML-models (Random Forest, Logistic Regression, Gradient Boosting Machine, Support Vector Machine, 3-layer Neural Network) and using different embeddings (average Word2Vec, Navec, Doc2Vec, FastText) of the news title, of its entire text, and of its first paragraph. We also considered the quality of the strategies of the mentioned ML models built on endogenous data (5 lags of the time series). We compare on our approach to SESTM model (Ke, Kelly & Xiu, 2020) that has shown promising results for the US stock market and English-language news and that, according to its authors, outperforms RavenPack algorithms (the industry-leading commercial vendor of financial news sentiment) in terms of Sharpe ratio scores. We show that our approach yields the best results more often than other included in the comparison. As topic modeling performs universally well across all European languages, our approach is expected to be applicable to all European stock markets, respectively.

The rest of the article is structured as follows. The ‘Related Work’ section reviews approaches based on interpretive sentiment analysis, methods based on combinations of embeddings and ML models, and topic models that are conceptually close to our framework. The ‘Methodology’ section introduces the proposed method. The ‘Datasets and preprocessing’ section describes the data used in the current study. The ‘Metrics’ section describes the return and risk metrics for portfolios obtained using various approaches discussed in this article. The ‘Experiments’ section contains a description of the procedure for forecasting and constructing various schemes for stock trend modeling. The ‘Numerical results’ section contains the results of our experiments. The ‘Discussion’ section interprets the obtained results. The ‘Conclusion’ summarizes our findings and discusses the possibilities for further framework improvements. Appendix A is devoted to a qualitative analysis of the results of topic modeling. This part of the article, first of all, compares the results of different topic models with each other. Second, it shows which topics are most frequently covered in the main federal media and in the trading terminal news. Finally, the temporal saturation of the market with new information is shown. Appendix B contains supplementary materials of this article, such as illustration of data and models, cumulative divergence of topic profiles, coherence scores and tables with the Granger causality test values.

Related Work

Much research exploring the relationship between textual information and financial time series relies on sentiment dictionaries, such as the Harvard-IV-4 dictionary and Loughran–McDonald Financial Dictionary (Loughran & Mcdonald., 2016). For instance, Li et al. (2014) use both of the mentioned dictionaries to create a sentiment-based model for stock market prediction tasks. Kim, Jeong & Ghani (2014) assign a sentiment score to a textual data stream using a dictionary and rules, after which the authors identify significance of correlations between this news stream and stock market fluctuations. Li et al. (2018) extract sentiment information using Loughran-McDonald, Harvard IV-4, and SenticNet 3.0 in their research. Picasso et al. (2019) use McDonald dictionary and AffectiveSpace 2 (Cambria et al., 2015) to evaluate sentiment information from financial news for twenty most capitalized companies listed in the NASDAQ 100 index. However, dictionary approach is hard to customize to specific data and prediction tasks. Existing dictionaries still require being extended to the financial domain. Moreover, sentiment dictionaries are still underdeveloped for less resourceful languages, including the Russian language and are the subject of recent research (Panchenko, 2014; Koltsova et al., 2020).

Other related works exploit various machine learning approaches (Rundo et al., 2019; Thakkar & Chaudhari, 2021) combined with different encoding procedures used to assign vector representations to documents; these procedures include TF-IDF features (Bing et al., 2017), word-embeddings (Mahmoudi, Docherty & Moscato, 2018) and deep learning methods (Matsubara, Akita & Uehara, 2018; Xu et al., 2020), among others. These vector representations, sometimes combined with other financial numerical features (Gu, Kelly & Xiu, 2020; Li, Wu & Wang, 2020) are used as an input for classification or regression models, depending on the time series problem statement (Henrique, Sobreiro & Kimura, 2019). For example, Khedr, S.E.Salama & Yaseen Hegazy (2017) propose the model that predicts the rise and fall of shares of companies traded at NASDAQ based on economic news. They combine stemming, n-gram, TF-IDF, and numerical features with Naïve Bayes and KNN algorithms. Manela & Moreira (2017) use n-gram features with the SVR model to estimate the relationship between the front-page text of The Wall Street Journal and the VIX volatility index. Weng et al. (2018) extract public information from Google and Wikipedia with Random Forest model (while simultaniously testing NN, SVR and boosted regression tree) to predict the 1-day ahead price of 19 additional stocks from different industries. Such approaches are difficult to interpret by a potential investor: it is often hard to understand why the vector representation model learned a certain word embedding and what effect it had on the final result, as well as to explain why the ML model chose a specific combination of non-transparent features as significant.

Latent Dirichlet Allocation (LDA) is one of the most popular topic modeling techniques using a Bayesian approach for generating topics (Blei, Ng & Jordan, 2003). Topics derived from topic modeling can be good predictors of financial time series. For instance, Chester Curme & Preis (2017) show that the forecasts of trading volume can be improved by accounting for news topical diversity which they measure as the Shannon entropy of a topic distribution yielded by a topic modeling algorithm run over daily corpora of Financial Times news. Also, there are natural extensions to the LDA model with the temporal structure of texts: DTM (Blei & Lafferty, 2006) and DIM (Gerrish & Blei, 2010). These models allow tracing temporal evolution of topics and their lexical composition and reveal the most influential documents. Other papers integrate text and time-series data into a single probabilistic model expanding DTM or LDA (Park, Lee & Moon, 2015; Kanungsukkasem & Leelanupab, 2019). In these papers, researchers carry out a qualitative analysis of topics associated with time series and evaluate the predictive power of the respective model. Kim et al. (2013) develop Topic Modeling with Time Series Feedback model (ITMTF) that infers topics iteratively while optimizing their correlation with time-series data in terms of its strength and direction. The latter means that topics are gradually re-defined so that to include only the words that affect the predicted time series in the same way (either negative or positive). This approach yields more causal topics than the baseline LDA in terms of Granger causality and more pure topics in terms of coherence of the effect’s direction. However, ITMTF model uses the time series data from a very limited pool of only three US companies, Apple and two airlines, and only for six months. Another important approach ideologically close to ours is SESTM—Sentiment Extraction via Screening and Topic Modeling (Ke, Kelly & Xiu, 2020). Its authors use a supervised topic model with two topics—one being assigned the words that have a positive impact on asset returns, and the other with the words having a negative impact—to calculate word-level predictive scores (termed sentiment scores) that are later transformed into text-level predictive scores. These latter scores are used to optimize investment portfolio construction whose quality is assessed in terms of Sharpe ratio and annual return metrics.

Methodology

We propose the Stock Tonal Topic Modeling approach (STTM) by introducing an index that reflects the association between topics occurring in news stream and the stock prices movement. This index, hereafter termed STTM index, is positive if the overall association of all the topics in the news stream of a given time period with the stock movement is positive (i.e., it predicts stock growth), and negative in the opposite case. Further, we use the STTM index to optimize investment portfolio construction and show its efficiency.

The proposed procedure of computing STTM index has several stages. First, we perform topic modeling of the news flow and calculate the salience of each topic at each time point of a pre-selected period, thus receiving a distribution of saliences for each topic over time. This distribution is further referred to as topic stream. Similarly, we obtain the word stream for each word as a distribution of word frequencies in our news flow over time. Second, we compute the tone of each word as the value of an association measure (in our case—Pearson correlation) between the word stream and the target time series (in our case—stock prices). Words found to be positively associated with the target time series are considered to have positive tone, and visa versa. Third, topic-level tone is computed based on the tones of high-probability words from each topic, according to a procedure described further below. Finally, topic-level tones are aggregated over all topics into a distribution termed tonal topic stream, which in turn is aggregated over time into a single value—STTM index. This index, thus, reflects the strength and the direction of the aggregated impact of all topics on the stock price movement.

More formally, let us denote textual data stream as collection D = (d1, td1), …, (dm, tdm), where di - document, tdi - date and time of document release. Let us also denote financial time series as pt = (p1, t1), …, (pN, tN), where pi - value, ti - corresponding time stamp. Our key problem is predicting ℙ(rt ≥ 0), where rt=pt−pt−1pt−1, with STTM index based on textual data flow between t and t − 1 as feature. In our notation, we normalize the raw STTM index to the range [0;1], so that STTM → 1 and STTM → 0 mean that the textual information pulls the time series pt up and down, respectively. We give a detailed description of the entire procedure whose graphical representation can be found in Fig. 1.

Figure 1 Stock Tonal Topic Modeling (STTM): framework structure.

Data preprocessing

First of all, we preprocess input text data. Each text is subject to tokenization and lemmatization, removal of stop-words, and punctuation symbols. Next, we calculate idf-parameter—inverse document frequency, part of TF-IDF feature, for each unique word w in D: (1) idfw,D= log|D||di∈D∣w∈di|,

where |D|—number of documents in collection; |{di ∈ D∣w ∈ di}|—number of documents from D collection, where the word w occurs. After that, we remove words in the upper and lower quantiles of the α-level from the text data.1 Thus, we do not consider the most rare or the most frequent words. Then, we transform the resulting text data into a bag-of-word representation.

Topic modeling

We feed the preprocessed text data as an input to the topic model for generating probabilistic topics: T1, …, Tn. The topic model can be LDA, DTM, DIM, ITMTF or any other technique.2 It can be pre-trained in advance or online-trained on the textual data stream D. As a result of topic modeling procedure, each document d is represented by n - dimensional vector of topics’ probabilities: θ(d) = (θd,1, …, θd,n). We numerically estimate the salience of each topic Tj in each time unit ti of the textual data stream D as follows: (2.1) Θij= ∑∀dintiθjd.

Topic stream TSj of each topic Tj is thus defined as a set of all salience scores of this topic in a given time period: (2.2) TSj=Θ0j,…,ΘNj.

Consequently, we associate each topic Tj with the time series of its stream TSj.

Topic tonality based on word-level tone

By analogy with topic stream, to define word stream, for each word w we first calculate its frequency as the sum of its occurrences over all documents d that have appeared in our stream in a given time point ti: (3.1) ciw= ∑∀dinticw,d.

Thus word stream WSw is defined as a set of word frequencies in a given time period: (3.2) WSw=c0w,…,cNw.

Consequently, we associate each word w with the time series of its stream WSw. In general, c(w, d) can be any additive function of the number of words w in the document d. The tone of the word w is determined as a function of target time series and word stream: (4.1) Ωw=fwpt,WSw.

Function fw can be any regression evaluation metric or any time series proximity metric. We use the Pearson correlation coefficient: (4.2) fw=rpt,WSw,

if the significance less than γ3 and (4.3) fw=0,

in other cases. Since each topic Tj is a probability distribution in each time unit ti over a dictionary V: Tj,i=w1,ϕw1j,i,…,w|V|,ϕw|V|j,i, we define the topic tone as a function of the word’s probabilities in the topic ϕWj,i and the corresponding word’s tone ΩW for each time point ti: (5.1) ψj,i=fTj,iϕWj,i,ΩW,

The overall tonality of topic Tj is defined as a set of its tones in a given time period: (5.2) ΨTj=ψj,0,…,ψj,N.

We implement function fTj,i as follows:

1. The number of the most probable words in the topic Tj at time point ti (i.e., the words for which the tone will be calculated) is selected so that the sum of their probabilities does not exceed the specified threshold Cj:4 (5.3) ∑wsorted by probabilityϕwj,i≤Cj,

2. The variables pProb and nProb are calculated. The calculations involve only words selected in the previous step: (5.4) pProb=∑Ωw≥0Ωwϕwj,i ∑|Ωwϕwj,i|,nProb=∑Ωw<0|Ωwϕwj,i|∑|Ωwϕwj,i|.

If the significance level rpt,WSw of all the selected words is higher than γ, then we define: (5.5) pProb=0,nProb=0.

The final topic tonality is expressed as the difference between pProb and nProb: (5.6) fTj,i=pProb−nProb.

Tonal topic stream

At the next step, we calculate the tonal topic stream (TTS) as a product of topic tonalities and topic streams for each time point: (6) TTS=Θ00⋅ψ0,0Θ10⋅ψ0,1…⋮⋱Θ0n⋅ψn,0ΘNn⋅ψn,N.

Thus TTS is a matrix, where rows correspond to the topics and columns correspond to the time points. Each element of this matrix, TTSj,i, is the value of tonal topic stream for the topic Tj at time point ti.

STTM index

Finally, STTM index as the overall tonality of all topics over a given period of time is defined as a time aggregate from the TTS matrix: (7) STTM=aggregatetTTS.

The aggregation function can be a simple or weighted mean, median, or sum. We use simple sum by default. The tonality of each specific news item d is, analogously, the aggregate over the products of topic probabilities of news item θ(d) and its topic tonalities ψ(d) at time point td. As noted above, for comparability purposes we normalize the STTM index to the range [0, 1] based on sigmoid regressor.

Figure 2 shows a general scheme for predicting directions of financial market movements using the STTM approach.

Figure 2 Scheme of the procedure for forecasting market movement based on STTM.

Datasets and Preprocessing

In this section, we describe in detail our datasets. We collect two types of data: financial time series data and textual data stream. We consider stocks included in the MOEX Russia Index as the time-series data and Russian-language news from the largest and most influential economic media as a textual data stream. In addition, we describe the required raw data preprocessing.

MOEX Russia Index

MOEX (Moscow Exchange) Russia Index (see Fig. 3) is a capitalization-weighted composite index serving as the primary ruble-denominated benchmark of the Russian stock market. It is calculated as the sum of the prices of 39 most liquid Russian stocks weighted by expert assessments of their impact on the Russian economy. These stocks are pre-selected by experts from among the largest and the most dynamically developing Russian issuers with economic activities in the leading sectors of the Russian economy. MOEX Russia Index is used as one of the baseline investment portfolios in this research. The shares time series from 2013 to 2021 included in MOEX Russia Index (available at: https://www.moex.com/en/index/IMOEX) listing constitute our times series dataset analyzed in this article.

Figure 3 Dynamics of MOEX Russia Index.

There is 39 main time series with the following tickers:

SBER, SBERP, GAZP, LKOH, YNDX, GMKN, NVTK, SNGS, SNGSP,

TATN, TATNP, ROSN, POLY, MGNT, MTSS, FIVE, MOEX, IRAO,

PLZL, NLMK, ALRS, CHMF, VTBR, RTKM, PHOR, TRNFP, RUAL,

AFKS, MAGN, DSKY, PIKK, HYDR, FEES, QIWI, AFLT, CBOM,

LSRG, RSTI, UPRO.

Each time series of the mentioned shares is converted to the weekly returns rt:

rt≡pt−pt−1pt−1,

where pt is the closing price of the shares time series, t - weekly timestamp.

Economic news

Our text dataset of daily news includes three Russian national media sources: Kommersant, RIA Novosti, and Vedomosti. Kommersant (The Businessman, available on the website: https://www.kommersant.ru) is a nationally distributed daily newspaper devoted to politics and business. RIA Novosti (Russian Information Agency, available on the website: https://ria.ru) is one of the principal state-owned news agencies publishing news and opinions of social, political, economic, scientific, and financial subjects. Vedomosti (The News, available on the website: https://www.vedomosti.ru) is a national daily newspaper specializing in business. In each media outlet we consider the texts from the economy section only. Consequently, it contains a significant number of editorials, analytical reviews, and expert opinions, affecting the estimated textual data flow. Table 1 illustrates the amount of collected data and the date intervals corresponding to it. Figure 4 shows the main statistics about the Kommersant newspaper. The top panel of the figure plots the histogram of the number of articles (ordinary news and analytical reviews) vs the number of characters. The bottom panel shows the distribution of the weekly number of articles over eight considered years. The drops in the number of news correspond to public holidays and weekends. The graphs for the other daily sources can be found in Figs. B1 and B2.

News preprocessing pipeline

We use a common natural language preprocessing pipeline. We begin with the tokenization5 by breaking each text into sentence components and then into word components. Next, we normalize all tokens in each article to lower case letters, remove punctuation, non-alphabetic, and non-Cyrillic symbols, and perform lemmaization with Yandex MyStem—an instrument developed specially for the Russian language and based on extensive morphological analysis.6 A lemmatizer was preferred to stemmers because it avoids aggressive suffix-stripping—an approach that would not be applicable for highly inflected languages, such as those of the Slavic family, since therein suffixes are heavily used for word formation and can entirely change word meaning. In terms of recognizing lemmas from word forms, Yandex MyStem has the error rate of about 2–3% only and consistently outperforms other existing models on different Russian-language corpora (Kotelnikov, Razova & Fishcheva, 2018). Finally, we remove stop words, such as prepositions, participles, interjections, numbers7 and the tokens in the upper and lower quantiles of the α-level idf-parameter. We use the standard 95%-quantile as a default value for α-level. Finally, we convert each news item into a vector of word counts.

Table 1 Summary statistics of collected daily news.

News agency name	Number of articles	Dates	
Kommersant	26, 132	01.01.2013–31.12.2021	
RIA Novosti	168, 285	01.01.2013–31.12.2021	
Vedomosti	3, 261	01.02.2015–31.12.2021	

Figure 4 Kommersant: The total number of articles by calendar week and the empirical distribution of the number of the symbols.

Metrics

We consider two approaches to assessing the results obtained. First, we explore the relationship between the stock market movement and news flow impact of the proposed tonal topic modeling procedure through correlation analysis using Granger’s causality test (Deveikyte et al., 2020). Second, we introduce a simple trading strategy of calibrating investment portfolios based on the predictions of the model and evaluate the performance of this strategy with the Sharpe ratio and the annual return of the portfolio (Ke, Kelly & Xiu, 2020). Economic metrics for evaluating the success of a calibrated portfolio appear to be more suitable for the stock trend prediction problem than standard classification metrics, such as accuracy, Receiving Operating Characteristics (ROC), and area under the curve (AUC). This is due to the following reasons. Standard classification quality metrics for one time series may not give high results. However, economic metrics work on many time series, so in this case we can get a good financial result.

Granger causality test

The Granger causality test8 is a statistical hypothesis test for determining whether one time series is useful in forecasting another. Granger causality requires time series stationarity. Let yt and xt be two time series. To see if xt ’Granger causes’ yt with maximum q time lag, the following regression is performed: (8) yt=a0+a1yt−1+...+aqyt−q+b1xt−1+...+bqxt−q.

Then, F-tests are used to evaluate the significance of the lagged x terms. The coefficients of lagged x terms estimate the impact of x on y. The null hypothesis that x does not Granger-cause y is accepted if and only if no lagged values of x are retained in the regression. Since in our case the time series has shown non-stationary behavior, as determined by the Dicky-Fuller unit root test,9 a first-order and, where necessary, a second-order differentiation was performed to achieve stationarity.

Sharpe ratio and annual return

The Sharpe ratio10 measures the performance of an investment, such as a share or a portfolio of shares compared to a risk-free asset, after adjusting for its risk. It is defined as the difference between the returns of the investment and the risk-free return, divided by the standard deviation of the investment returns. It represents the additional amount of return that an investor receives per unit of increase in risk. More formally, (9) Sharpe ratio=Rp−Rfσp,

where Rp - return of portfolio, Rf - risk-free rate, σp - standard deviation of the portfolio’s excess return. As a risk-free asset for the Russian market, we use government bond yields (available on the website: https://www.cbr.ru/eng/hd_base/zcyc_params/). A Sharpe ratio of less than one is usually considered unacceptable or bad. It means that the risk of portfolio encounters is being offset well enough by its return.

Annual return or Compounded Annual Growth Rate (CAGR)11 is the average annual rate calculated from the returns observed in the first and the last years of a given time span, assuming that all the dividends are reinvested in the end of each year. More formally, (10) Annual return=1+EV−SVSV1n−1,

where EV - ending value, SV - start value, n - number of years.

Experiments

In this section, we describe our experimental procedures. The ‘STTM’ subsection describes how to build and evaluate the quality of models based on the proposed framework. The ‘SESTM’ subsection describes how to build a topic model that outperforms RavenPack, the industry-leading commercial vendor of financial news sentiment. The ‘Shallow feature based methods of text processing’ subsection describes a scheme for building models based on embedding news. The ‘Endogen models’ subsection reveals a way to build simple endogenous models on time series lags. The ‘Evaluation procedure’ subsection describes the scheme for splitting the initial data into training and test samples to evaluate quality metrics.

STTM

As mentioned before, our Stock Tonal Topic Modeling (STTM) approach can have any basic topic model as its core. In this article, we implement two models: LDA12 and DTM13 with python-wrapper from gensim python-package: https://radimrehurek.com/gensim/. Qualitative analysis of these models is presented in the section ‘Qualitative Analysis of Russian Economic News Topic Modeling’ in Appendix A. DTM extends LDA by allowing word probabilities in a topic to change over time. We have chosen to update them in the increments of one month. Further, we select the number of topics so as to avoid solutions with either a large number of too granular topics or a small number of too broad topics. To do so, we optimize topic coherence with Roder’s Cv metric (Röder, Both & Hinneburg, 2015). Although the dynamics of coherence change with the increase of the number of topics is somewhat different for different news sources (see Figs. B4, B5 and B6), the overall optimum appears at n = 20, where Cv curve reaches its maximum before flattening out for all national media sources. This optimum is the same for both LDA and DTM. We run a topic model on the training dataset and apply it to the test dataset. The datasets are constructed according to the procedure described in ‘Evaluation procedure’ section below. Topic tonality based on word-level tone is calculated from the solution obtained on the training set. Topic stream is calculated from the the test set, and topic tonality based on word-level tone is applied to it. STTM hyper-parameters are selected based on optimization of ROC-AUC metrics the grid-search on the training set for each time series independently. Examples of the STTM index, the stock time series, news, topics, words and also their tonalities and tones, as well as an example of a tonal topic stream are presented in Figs. B9, B10, B11 and B12. It can be seen that results of proposal procedure are highly interpretable. Since topic modeling possesses a certain level of instability leading to fluctuations in the word probabilities, we repeat all calculations for trading strategy performance at least ten times. After that we estimate the mean and variance of each of the considered economic metrics.

SESTM

We implemented the Sentiment Extraction via the Screening and Topic Modeling (SESTM) procedure (Ke, Kelly & Xiu, 2020) as a baseline topic model. We apply it for each time series from the MOEX Russia Index for three national news sources. SESTM approach infers only two topics—containing words that have either negative or positive effect on the target time series, the composition of these topics being optimized iteratively based on an association metric. The process consists of three steps: 1. isolating a list of sentiment terms via predictive screening 2. assigning sentiment weights to these words via topic modeling 3. aggregating terms into an article-level sentiment score via penalized likelihood. The main assumption of the model is that the news is generated from the following mixture multinomial distribution: (11) di,S∼Multinominalsi,piO++1−piO−,

where si is the total count of sentiment-charged words in article di, pi is the sentiment score, O+ and O− are a positive and negative topics, respectively, which is probability distributions over words. The objective of SESTM procedure is to learn the model parameters O+, O− and pi. SESTM algorithm has three hyper-parameters of screening for sentiment-charged words and one hyper-parameter for regularization in learning-optimization problem. All hyper-parameters are tuned through the time cross-validation procedure with l1-norm of the differences between estimated article sentiment scores and the corresponding standardized return ranks as a loss function. (Figures B13 and B14) in Appendix B contain examples of the most common words with cumulative positive (negative) tones, and the most cumulatively positive (negative) tonal words for VTB (VTBR) share prices. It can be seen that the contents of these topics is mixed and broadly uninterpretable.

Shallow feature based methods of text processing

To compare our approach to the models with shallow features based methods of text preprocessing, we apply a specific pipeline shown on Fig. 5. For each economic news from the considered news agencies (Kommersant, Vedomosti, RIA Novosti), various textual components are extracted (full text, first paragraphs, titles). After that, each component is preprocessed as described in section ‘Datasets and preprocessing’. Further, various techniques for obtaining embeddings are applied to each news item: Word2Vec, Navec, Doc2Vec, FastText,14 Navec realization from natasha python-package: https://github.com/natasha/navec, FastText realization as python-package from: https://fasttext.cc/ (all embeddings models trained on the first two years of texts for each news outlet separately). The resulting embeddings of news (where all the news for the same week are treated as one document) are fed as the input features to the following machine learning models: Random Forest (RF), Logistic Regression (LR), Gradient Boosting Machine (GBM), Support Vector Machine (SVM), and 3-layers Neural Network (NN).15 The target variable for all models is the sign of returns for each ticker included in MOEX Russia Index, which equals 1 for the growing times series and 0 otherwise.

Figure 5 Pipeline for constructing shallow feature based methods of text processing.

Endogenous models

In addition, we compare the proposed framework with simple endogenous models: Random forest (RF), logistic regression (LR), gradient boosting machine (GBM), support vector machine (SVM), and three-layers neural network (NN), where weekly return lags are used as features, and the target variable is the same as in shallow-feature-based models. Figure 6 shows the pipeline we used for construction of such endogenous models.

Figure 6 Pipeline for constructing endogen models.

Evaluation procedure

As for time series data, test and train datasets have to be defined as subsets covering uninterrupted periods of time, we use an iterative expanding cross-validation scheme (visualised in Fig. 7) in which we expand the time window of the training set by one year at each iteration, starting from two years and ending with six years, while test set window is kept stable at the length of one year across all iterations. Topic model obtained on the training set is retrained at each iteration and then applied to the test set. We estimate our models on all economic news between stock market start time each Monday and its end time each Friday. Considered target variable is movement direction sign(rt) between the closing price on Friday and the opening price on the previous Monday for each share in the MOEX Russian Index.

Figure 7 Train-test splitting: expanding window scheme.

For the proposed STTM approach, we compute a Granger causality test between the weekly value of STTM index and the weekly stock price of each ticker included in MOEX Russia Index for each test interval separately. For each ticker and for different topic models (LDA and DTM), including STTM, we evaluate the weekly trading strategy performance. We use a straightforward long strategy. For this trading strategy each Friday at the time of the closure of the stock market we select top 20 percent of stocks with the highest value of the model prediction for the current week. These stocks are the most likely to demonstrate price gains in the upcoming week. Next, we buy these 20% of stocks at the Friday prices. Such procedure is repeated each Friday thus providing portfolio recomposition. We then evaluate this strategy with the annual returns and Sharpe ratio metric introduced before. In doing so, we do not account for either broker commission or transactions costs (see Limitations section) because our goal is to evaluate the predictive power of our method as compared with other methods, rather than to calculate the amount of the final return it allows to gain.

Numerical Results

Granger causality tests

In this section, we provide numerical results for the Granger causality test between STTM index and Friday’s prices for each of 39 tickers included in MOEX Russia Index; this is done for two topic models employed (LDA and DTM) and for three different news sources. Calculation details can be found in Tables C1–C6. Figure 8 shows the proportion of the assets in our sample for which the STTM index has significant Granger predictive power in each of the studied years.

Weekly trading strategy performance

In total, we compare the performance of 28 different portfolios: five endogenous model based portfolios, 20 portfolios using shallow feature based methods of text processing, two portfolios based on the proposed STTM approach, and one based on SESTM. Each of these mentioned portfolios is constructed for three different news sources independently. Given that we have 39 tickers in our analysis, in total we obtain 2,886 different models validated on six train/test splits each. We also compare all these models to two baselines: MOEX Russia Index, as a type of a broad stock market index, introduced above, and Equal Weight Index (EWI) based on MOEX’s tickers, as a type of buy and hold strategy. While the former exemplifies capitalization-weighted index, the latter gives equal weight to all stocks, including small-cap stocks that are generally considered to be higher risk and to have higher potential return investments compared to large-caps. In theory, giving greater weight to the smaller names of the MOEX Russia Index in an equal-weight portfolio should increase the return potential of the portfolio, so EWI may be expected to perform better than MOEX Russia Index.

Table 2 contains information about performance of the baselines. Table 3 demonstrates the results for topic modeling based approaches, Table 4 contains the results for shallow feature based methods of text processing and Table 5 presents the results for the endogenous models.

Figure 8 Number of significant Granger correlations as a percentage of portfolio size for different models and news sources.

Figure 9 shows how weekly returns accumulate depending on the model used for forming an investment strategy and compares them to MOEX Russia Index (IMOEX) and Equal Weight Index (EWI) as baseline strategies. Each strategy uses one-week-ahead approach and sorts portfolios by the score obtained from the chosen model. Specifically, Figs. 9A–9C plot portfolio return accumulation graphs for the strategies based on one of the eight models (STTM (LDA), STTM (DTM), SESTM approaches and five best returns of strategies built on shallow feature based methods of text processing). Each facet A-C presents strategies using the data from only one information source: Kommersant, RIA Novosti and Vedomosti, respectively. Figure 9D contains the best models that have a Sharpe Ratio greater than one. We discuss it in detail further below.

Table 2 Trading strategy performance for economics baselines.

Economic baseline	Sharpe ratio	Annual return	
Equal Weight Index (MOEX)	0.7311	0.1965	
MOEX Russia Index	0.4377	0.1851	
Equal Weight Index (MOEX) from 2017	0.2626	0.1038	
MOEX Russia Index from 2017	0.2391	0.1032	

Discussion

In this section we interpret the obtained results.

Granger causality tests

Figure 8 shows that, as the time passes and the volume of the training data increases, the proportion of tickers for which our STTM approach turns to be Granger-causal tends to increase as well. An exception is a sharp fall of the predictive power for the majority of models in 2018. This fall is most probably explained with a number of international macroeconomic events in the second half of 2018 (including USA-China trade wars, US Federal Funds Rate hike, and the collapse of a large number of global financial indices). These events were poorly covered in the Russian media which focused on the internal agendas, such as the resonant raise of the retirement age. From Fig. 8 it can be seen that, according to the Granger causality test, the proposed STTM index can be significant for as much as 70% of stock quotes listed in the MOEX Russia Index if the data is sufficient to calibrate the model.

Table 3 Trading strategy performance for topic modeling approaches: STTM (LDA), STTM (DTM) and SESTM for Kommersant, Vedomosti and RIA Novosti economic news sources.

Values of the Sharpe metric greater than one are marked in bold.

Kommersant	
STTM (LDA)	STTM (DTM)	SESTM	
Sharpe ratio	Annual return	Sharpe ratio	Annual return	Sharpe ratio	Annual return	
1.3706 ± 0.0907	0.3612 ± 0.0212	1.0798 ± 0.0653	0.2845 ± 0.0201	0.4830	0.1598	
Vedomosti	
STTM (LDA)	STTM (DTM)	SESTM	
Sharpe ratio	Annual return	Sharpe ratio	Annual return	Sharpe ratio	Annual return	
1.1059 ± 0.0665	0.2840 ± 0.01853	0.7941 ± 0.0452	0.2095 ± 0.0319	0.3370	0.1296	
RIA Novosti	
STTM (LDA)	STTM (DTM)	SESTM	
Sharpe ratio	Annual return	Sharpe ratio	Annual return	Sharpe ratio	Annual return	
1.0140 ± 0.0572	0.2755 ± 0.0157	0.9691 ± 0.0570	0.2675 ± 0.0183	0.6433	0.2050	

Table 4 Trading strategy performance for shallow feature based methods of text processing constructed on the full texts, title and the first paragraphs of news from Kommersant, Vedomosti and RIA Novosti economic news sections. Top five values of the Sharpe ratio and annual return for each news agency are marked in bold.

Kommersant	
Scheme		Text	Title	
	Sharpe ratio	Annual return	Sharpe ratio	Annual return	Sharpe ratio	Annual return	
GBM + Doc2Vec	0.3221	0.1301	0.9141	0.2614	0.3199	0.1294	
GBM + FastText	0.7707	0.2173	0.5149	0.1675	0.3854	0.1419	
GBM + Navec	0.471	0.1612	0.6731	0.1994	0.4041	0.1457	
GBM + Word2Vec	0.465	0.1616	0.5807	0.1806	0.1637	0.0964	
LR + Doc2Vec	0.516	0.1669	0.6622	0.1991	1.0056	0.2746	
LR + FastText	0.7131	0.2059	0.9137	0.2529	0.8933	0.2507	
LR + Navec	0.4284	0.1514	0.4553	0.1564	0.5462	0.1785	
LR + Word2Vec	0.5052	0.1644	0.8237	0.2329	0.586	0.1867	
NN + Doc2Vec	0.5267	0.1731	0.2439	0.1128	0.8192	0.236	
NN + FastText	0.7926	0.2278	0.401	0.1449	0.6687	0.2052	
NN + Navec	0.551	0.1767	0.5506	0.1719	0.44	0.156	
NN + Word2Vec	0.6792	0.1994	0.3565	0.1356	0.401	0.1449	
RF + Doc2Vec	0.0502	0.0747	0.6528	0.2056	0.6221	0.1894	
RF + FastText	0.6838	0.2022	0.4175	0.1493	0.499	0.1641	
RF + Navec	0.7199	0.2137	0.1372	0.0919	0.3583	0.1355	
RF + Word2Vec	0.3072	0.1262	0.4723	0.1578	0.203	0.1047	
SVM + Doc2Vec	0.5142	0.1742	0.6416	0.1998	0.6	0.1895	
SVM + FastText	0.1371	0.0911	0.551	0.1815	0.2911	0.1246	
SVM + Navec	0.5364	0.175	0.618	0.1933	0.3577	0.1385	
Vedomosti	
Scheme	Text	Title	First paragraph	
	Sharpe ratio	Annual return	Sharpe ratio	Annual return	Sharpe ratio	Annual return	
GBM + Doc2Vec	0.1629	0.0825	−0.0552	0.0375	0.7428	0.2093	
GBM + FastText	−0.0522	0.0397	0.2586	0.1025	0.2487	0.1004	
GBM + Navec	0.7793	0.2143	0.216	0.0941	0.7344	0.2096	
GBM + Word2Vec	0.0747	0.064	−0.3653	−0.0164	0.3274	0.1161	
LR + Doc2Vec	0.3098	0.1149	0.564	0.1672	0.1519	0.0806	
LR + FastText	0.1202	0.0709	0.1435	0.0794	0.2406	0.1001	
LR + Navec	0.5791	0.1789	0.3249	0.1198	0.3092	0.1145	
LR + Word2Vec	0.2298	0.0974	0.3287	0.1168	0.5277	0.166	
NN + Doc2Vec	0.2259	0.0954	0.8721	0.2384	0.2487	0.1018	
NN + FastText	0.2525	0.1024	0.3038	0.1133	0.1523	0.0798	
NN + Navec	0.7775	0.2111	0.5409	0.1661	0.3324	0.1207	
NN + Word2Vec	0.1523	0.0798	0.3735	0.1266	0.1523	0.0798	
RF + Doc2Vec	0.0259	0.054	0.5677	0.1704	0.6729	0.1908	
RF + FastText	−0.209	0.0048	0.1558	0.0815	−0.2493	0.0052	
RF + Navec	0.449	0.1437	0.345	0.1264	0.3384	0.1191	
RF + Word2Vec	0.0329	0.0535	0.2996	0.1114	0.1973	0.0898	
SVM + Doc2Vec	0.4038	0.1384	0.1014	0.0681	0.0764	0.0654	
SVM + FastText	0.3401	0.1186	0.5436	0.1643	0.2105	0.0929	
SVM + Navec	0.0825	0.0668	−0.0008	0.0475	0.3267	0.1183	
RIA Novosti	
Scheme	Text	Title	First paragraph	
	Sharpe ratio	Annual return	Sharpe ratio	Annual return	Sharpe ratio	Annual return	
GBM + Doc2Vec	0.8347	0.2414	0.9204	0.2653	0.6241	0.1953	
GBM + FastText	0.8641	0.2454	0.8971	0.2624	0.3022	0.1264	
GBM + Navec	0.3561	0.1382	0.3561	0.1382	1.1917	0.3284	
GBM + Word2Vec	0.71	0.2172	0.71	0.2172	0.602	0.1912	
LR + Doc2Vec	0.2994	0.126	0.2994	0.126	0.8791	0.2538	
LR + FastText	0.4763	0.1633	0.4763	0.1633	0.8888	0.2655	
LR + Navec	0.401	0.1485	0.401	0.1485	0.7374	0.2227	
LR + Word2Vec	0.4652	0.1612	0.4652	0.1612	0.5586	0.183	
NN + Doc2Vec	0.2146	0.1074	0.2034	0.1051	0.4429	0.1541	
NN + FastText	0.4065	0.149	0.4065	0.149	0.5885	0.1857	
NN + Navec	0.427	0.1518	0.427	0.1518	0.5871	0.19	
NN + Word2Vec	0.3289	0.1317	0.3289	0.1317	0.5885	0.1857	
RF + Doc2Vec	0.5385	0.1806	0.8189	0.2462	0.7877	0.2307	
RF + FastText	0.3982	0.1449	0.3982	0.1449	0.9746	0.2831	
RF + Navec	0.3774	0.1409	0.3774	0.1409	0.5307	0.1798	
RF + Word2Vec	0.2479	0.1144	0.2479	0.1144	0.2759	0.1205	
SVM + Doc2Vec	0.5084	0.1704	0.5055	0.1698	0.7373	0.2166	
SVM + FastText	0.3764	0.1434	0.3764	0.1434	0.611	0.1944	
SVM + Navec	0.431	0.1535	0.431	0.1535	0.4775	0.1594	

Table 5 Trading strategy performance for approaches based on five-lags endogenous models.

Endogen model	Sharpe ratio	Annual return	
GBM	0.7125	0.2135	
LR	0.6010	0.1851	
NN	0.6347	0.1952	
RF	0.4693	0.1617	
SVM	0.6213	0.1904	

Figure 9 Out-of-sample cumulative returns of one-week-ahead strategies.

(A–C) returns of strategies based on eight best models and two baseline models that use data from Kommersant, RIA Novosti and Vedomosti, respectively. (D) strategies with Sharpe-ratio more than one and MOEX Russia Index & Equal Weight Index as a baselines.

Weekly trading strategy performance

The D facet of Fig. 9 illustrates one-week-ahead performance of most economically successful portfolios (Sharpe ratio more than one), as well as the MOEX Russia Index. The top models in terms of the success of the investment portfolio built on them are as follows: STTM (LDA) on Kommersant with mean Sharpe ratio 1.3706 (annual return 36.12%), GBM + Navec based on RIA Novosti first paragraph with Sharpe ratio 1.1917 (annual return 32.84%), STTM (LDA) on Vedomosti with mean Sharpe ratio 1.1059 (annual return 28.40%), STTM (DTM) on Kommersant with mean Sharpe ratio 1.0798 (annual return 28.45%), STTM (LDA) on RIA Novosti with mean Sharpe ratio 1.0140 (annual return 27.55%), and LR + Doc2Vec based on Kommersant’s titles 1.0056 (annual return 27.46%). Portfolios built on the models mentioned above are also the most profitable. So out of 69 different approaches to stock trend prediction, only six turned out to be economically viable. Among them, four are built on our novel proposed approach STTM and only two are derived using shallow feature based text processing methods. Three out of six are based on Kommersant news agency data, two are based on RIA Novosti data, one is based on Vedomosti.

Now let us take a closer look at the best models for each individual news agency. A, B, C facets of Fig. 9 display performance for news sources Kommersant, RIA Novosti and Vedomosti, respectively. Each facet includes all topic modeling based approaches and five most successful models out of the remaining approaches (shallow feature based methods of text processing and endogenous models). For the Kommersant news agency, the STTM (LDA), model takes the first place (Sharpe ratio 1.3706 and annual return 36.12%), followed by STTM (DTM) (Sharpe ratio 1.0798 and annual return 28.45%) both in terms of Sharpe ratio and annual return. Next are five models with shallow feature based methods of text processing. For RIA Novosti STTM (LDA) and STTM (DTM) take the second (Sharpe ratio 1.0140 and annual return 27.55%) and the fourth (Sharpe ratio 0.9691 and annual return 26.75%) places respectively, the rest of the best models are shallow feature based methods of text processing. For Vedomosti STTM (LDA) and STTM (DTM) take the first (Sharpe ratio 1.1059 and annual return 28.40%) and third (Sharpe ratio 0.7941 and annual return 20.95%) places, respectively, the remaining best models again are shallow feature based methods of text processing.

Our experiments show that for all news sources the proposed STTM approach is among the best models, while maintaining the interpretability of results (see Figs. B9, B10, B11 and B12 in Appendix B). It is worth noting that endogenous models do not make it to the top of the best models, which in turn indicates that more useful economic information can be obtained from external data sources as compared to the information contained in the time series. SESTM never gets in any list of the best strategies, possibly, due to a smaller size of our dataset as compared to the dataset used by SESTM developers. However, we note that SESTM still outperforms the general economic baseline MOEX Russia Index both in terms of Sharpe ratio and annual return.

Conclusion

In this article, we have proposed a new approach—STTM—for evaluating the impact of news stream on the stock market trend, which is novel in several aspects. First, it does not use domain-specific dictionaries or any other manual markup. Next, unlike many commercial solutions, such as Reuters and Bloomberg, which produce general impact coefficients for the entire market, our algorithm can be fine-tuned for any individual issuer. At the same time, our analytical pipeline remains transparent and interpretable for an investor or a risk manager. It clusters news streams via topic modeling, finds the most influential terms among the most probable words of each topic with a tone assessment procedure, and offers assessment of the overall tone of each topic through trade-off between positive and negative terms and their probabilities, as well as tone aggregation across the entire news stream. Topic tone reflects the strength and the direction of its potential impact on stock prices. Our procedure can be combined with various topic modeling techniques and time series proximity measures. It can also be generalized to other domains and used to assess the impact of text data on a various time series, both in predictive or explanatory tasks.

To illustrate the usefulness of the proposed method, we have carried out a large number of experiments on the prediction of the Russian stock market with the texts from the economic sections of the most significant Russian-language news editions. We investigated Granger causality between the output of the proposed STTM approach and each of the 39 tickers included in the MOEX Russia Index for six years and for two different topic modeling algorithms (LDA and DTM). The model shows significant causality across multiple tickers and can Granger-cause more than 70% of those if the training data is large enough. We compared 28 different models by assessing their performance in terms of efficiency of a simple long-term trading strategy. For that, we created portfolios based on the predictions from each of these models and from each of our three news sources independently: 20 portfolios used shallow feature based methods of text processing, one was based on SESTM, five on endogen models, and two our approach (STTM). This corresponds to the construction of 2,886 different model variations, as each of the portfolio creation method was applied to each of the 39 tickers and on validated on six train/test splits. The quality of the resulting portfolios was evaluated by two metrics: Sharpe ratio and annual return.

Of all the multitude of model variations, only six turned out to be economically viable with Sharpe ratio more than one. Of them as many as four were based on STTM, and the remaining two were shallow feature based text processing methods that were initially represented by a much large number of model variations than STTM. Each of the STTM-based models ranked top of the list for various news publications, consistently outperforming the MOEX Russia Index baseline, the endogenous models, and the SESTM-based topic model. Thus, our work shows that the proposed framework is promising in explaining and predicting financial time series based on the textual data flow. The universal applicability of topic modeling to all European languages, as well as to some other languages, allows to assume that this framework has good prospects of being usable far beyond the Russian stock market.

The novelty of STTM, as compared to other approaches that make use of topic modeling- SESTM (Ke, Kelly & Xiu, 2020) and ITMTF (Kim et al., 2013)—is two-fold. First, STTM allows to directly optimize the efficiency of investment portfolios—a task that ITMTF does not address—and does it better than SESTM. Second, both SESTM and ITMTF work to homogenize the generated topics by the direction of their effect on the target variable—either negative and positive. For this purpose, SESTM reduces the number of topics to two only which renders them uninterpretable (and, as we have shown, less predictive than our approach). ITMTF’s approach is more nuanced: while optimizing both topics’ predictive power and their purity in terms of the effect’s direction, it yields really interpretable topics. However, it does not evaluate the overall effect of the entire news stream of a given time period on the share prices which, ultimately, is the main practical goal of using news in such models. Additionally, it is not obvious that the overall predictive power of purified topics is higher than that of naturally occurring topics. Thus, adaptation of ITMTF purification logic to the goal of direct trading strategy optimization and comparison of the resulting pipeline to STTM is a an interesting task for future research.

Our approach has several practical implications. First, its ability to create impact indices of a news stream or a stream of textual data from social media for an individual issuer should be of higher practical value for traders than overall market indices. Issuer-specific indices can be used directly in trading strategies or as a factor in more complex models. Second, transparency and interpretability of our approach should make it attractive to investment applications for the mass user that are appearing on the market in large numbers. Our approach can make decision advices rendered by such apps more understandable for lay investors and thus increase customer trust and loyalty to such apps. Finally, professional risk analysts can benefit from the in-depth analysis of the rich information provided by our approach. They can numerically analyze the behavior of their companies in the past for better risk management in the future.

Limitations

Like all approaches involving topic modeling, our approach is sensitive to duplicate news. Although a large amount of duplicates may indicate topic’s importance, duplicate-based topics tend to be artificially separated from similar, but not identical texts. The effect of this phenomenon on model performance needs to be studied experimentally. Likewise, coverage of economic events may be heavily skewed by editorial choices that, like in 2018, may hinder model’s predictive power. This effect might be mitigated by broader samples of media outlets. Finally, as it was mentioned, in this article we ignore brokers’ commissions and transaction costs when evaluating the performance of our strategy. Although here our goal is to find return predictive signals, for models aiming at exact calculation of returns’ amounts these additional costs should be accounted for.

Supplemental Information

Supplemental Information 1 Time series rows

Click here for additional data file.

The study was implemented in the framework of the Basic Research Program at the National Research University Higher School of Economics (HSE University) in 2022.

Appendix A

In this supplementary section, we conduct a qualitative analysis of various topic modeling techniques, relationships between general economic news and trading terminal news.

Trading terminal news

We consider real-time trading terminal news produced by the Interfax agency (financial and economic news product, trial available on the website: https://interfax.com/products/news-products/). We collected 739,680 news items from Jan 1, 2017 to Jan 1, 2020. Graphs of articles, total number by calendar week, and the empirical distribution of the number of the symbols are in supplementary materials (Fig. B3). Note that the number of articles per week correlates with similar charts for the general economic news. It illustrates the shared imagination of economic processes in both general economic and trading news. We can also note the lack of analytical review among trading news. Within the day, trading news is distributed in one modality. Figure A1 plots the average number of articles in each hour interval of the day.

Qualitative analysis of russian economic news topic modeling

We investigated the similarity of national news agencies and estimated the amount of new information contained in the trading terminal news in terms of topic modeling. We train topic modeling algorithms on each of the national media sources. After that, we apply pre-fitted models to real-time news from the trading terminal. The choice of this order is due, on the one hand, to the technical features of topic modeling algorithms: in longer texts, it is easier to highlight topics, and on the other hand to the natural features of the editorial policy: in national media, the news is published regarding the appropriate context, while trading news is published as is, and contains a lot of irrelevant noise. We use two baseline topic models: LDA and DTM. As noted in subsection ‘STTM’ of the section ‘Experiments’, we set one month as the time interval for the change in the word’s probabilities in the DTM’s topic and the number of topics n = 20 that gave the highest CV score, before the CV-score graph flattening out for all national media sources. The resulting topics can be titled as follows: macroeconomic indicators, Central Bank statements, pension legislation, tax law, economic reforms, monetary policy, financing of national projects, public procurement, trade duties, investment climate and economic development, export figures, rules in entrepreneurship and trade, energy tariffs, insurance, digital technologies, international trade agreements, debt burden, labor and employment, mining and energy, and international relations. Figure A2 shows the topic similarities for different models and data sources, aligned using the Hungarian algorithm.16 Since the distribution of words in the DTM model topics varies from month to month, we use the time-averaged distribution of words for each considered topic. The topic modeling results show high cosine similarity between the same models based on data from different sources, and between different models based on the same sources. The exception is the national agency Vedomosti, the LDA model of which is poorly consistent with the DTM model and the LDA models, based on the data from Kommersant and RIA Novosti. Further, we apply the obtained models to real-time news from the trading terminal.

10.7717/peerjcs.1156/fig-A1 Figure A1 The average numbers of articles per hour (24 h EST time) from Jan 1, 2017 to Jan 1, 2020.

10.7717/peerjcs.1156/fig-A2 Figure A2 Cosine similarities between the topics from different models (LDA, DTM) and national media (Kommersant, Vedomosti, RIA Novosti): each row and column correspond to the topic indexes and color strength indicates the cosine similarity.

We can estimate the amount of new information in intraday data through the diversity feature (Chester Curme & Preis, 2017), which characterizes the topic model’s degree of confidence: (12) Ht≡−∑n=1Nρt,n logρt,n,

where ρt,n is the relative weight of topic n in the news on time interval t. Our hypothesis is simple: each news item should belong to a small number of noticeable topics. Respectively, the diversity feature of the topic model should be minor. When there are no suitable topics for the news in the topic model, it strives to distribute the probabilities evenly. Thus the diversity indicator is overestimated. Figure A3 shows the LDA model’s diversity distributions in its training sample (Kommersant) and applied to real-time news. We can observe a slight shift in the distribution of diversity, but the model shows significant confidence in general. From these considerations, we can estimate the amount of new information contained in trading terminal news and, at the same time, is lost in the general economic news. Also, topic streams (general economic news vs. trading news) show a significant correlation (see Fig. B8), which once again confirms the unity of the described economic processes. On the other hand, there is a cumulative divergence of topic profiles (see Fig. B7): national media tend to write on the following topics: macroeconomic indicators, Central Bank statements, pension legislation, whereas real-time news write more often about: labor and employment, mining and energy, international relations.

10.7717/peerjcs.1156/fig-A3 Figure A3 The density of diversity feature for real-time news from the trading terminal produced by Interfax and daily news from Kommersant national media: estimation of new information contained in intraday news and is lost in daily news.

In addition, we estimated the distribution of the received topics within the considered day. Figure A4 shows the cumulative division of the intraday news into topics: every hour, we calculate the total topics probabilities of new incoming data, add it to the amounts already received for the previous hours, and normalize. You can see the saturation of topics from a certain hour in the figure. To determine the elbow point on the timeline, we use the KL divergence function between the cumulative topic distribution for a specific time and the final topic distribution at the end of the day: (13) DKLPt||Q= ∑i=1nρt,i logρt,iqi,

where ρt,i is the cumulative weight of topic i in the intraday trading news on time interval t and qi is final weight of topic i at the end of the day. After that, we determine when the graph of the above-described KL divergence function (middle graph of Fig. A4) reaches a plateau and find the elbow point. We use Kneedle algorithm17 for this purpose. In the given example (middle graph in Fig. A4), topic saturation occurs from 8 a.m. Since that time, the picture of the profile of the topics almost does not change within the day, and new information basically clarifies the previous. We performed the above procedure for all dates in the dataset of intraday trading news and obtained the following picture of the distribution of topics profiles saturation points. This is demonstrated in Fig. A5. In the figure, we have one pronounced data modality with a center at 9 a.m. It is consistent with the opening time of the Moscow Exchange. Thus the main discussion of the economic situation in the Russian trading news takes place before the start of trading.

10.7717/peerjcs.1156/fig-A4 Figure A4 The upper graph illustrates the cumulative distribution of topics in real-time trading news for a particular day.

The middle graph shows the KL-divergence of the cumulative distribution of topics at a specific time and the end of the day. The lower chart shows the total number of new news at a particular time.

10.7717/peerjcs.1156/fig-A5 Figure A5 The distribution of topic profiles saturation points.

Appendix B

News data figures

10.7717/peerjcs.1156/fig-B1 Figure B1 Vedomosti: The total number of articles by calendar week and the empirical distribution of the number of the symbols.

10.7717/peerjcs.1156/fig-B2 Figure B2 RIA Novosti: The total number of articles by calendar week and the empirical distribution of the number of the symbols.

10.7717/peerjcs.1156/fig-B3 Figure B3 Interfax: The total number of articles by calendar week and the empirical distribution of the number of the symbols.

10.7717/peerjcs.1156/fig-B4 Figure B4 Kommersant: Dynamics of the CV-measure score.

Topic Modeling: CV score

10.7717/peerjcs.1156/fig-B5 Figure B5 Vedomosti: Dynamics of the CV-measure score.

10.7717/peerjcs.1156/fig-B6 Figure B6 RIA Novosti: Dynamics of the CV-measure score.

Topic Modeling: national media and trading terminal news comparison

10.7717/peerjcs.1156/fig-B7 Figure B7 Cumulative divergence of topic profiles between Kommersant and Interfax news agencies.

The left side of the figure: topics with more significant topic stream in Interfax. The right side of the figure: topics with more significant topic stream in Kommersant.

10.7717/peerjcs.1156/fig-B8 Figure B8 Distribution of Pearson correlation p-value between topic streams in Kommersant and in Interfax.

STTM index: illustrations

10.7717/peerjcs.1156/fig-B9 Figure B9 Illustration of STTM index.

The right figure shows the dynamics of the share price, on which positive values (green), negative values (red) and near-zero values (blue) of the raw STTM index are plotted in color, as well as economic news sorted by their tonality for the selected date. The upper left figure shows the topics and their tonality for a given news, as well as the distribution of words in the topics and the tones corresponding to these words. The lower left figure shows the tonal topic stream (TTS).

10.7717/peerjcs.1156/fig-B10 Figure B10 The dynamics of the share price with positive values (green), negative values (red) and near-zero values (blue) of the raw STTM index, as well as economic news sorted by their tonality for the selected date.

10.7717/peerjcs.1156/fig-B11 Figure B11 Illustration of Tonal Topic Stream (TTS).

10.7717/peerjcs.1156/fig-B12 Figure B12 Topics and their tonality for a given news, as well as the distribution of words in the topics and the tones corresponding to these words (red color correspond to negative, green color correspond to positive, blue color correspond to zero tonality/tones of the topics/words).

SESTM Model: frequency and tonality

10.7717/peerjcs.1156/fig-B13 Figure B13 Left: most cumulative positive tonal words, Right: most common words with cumulative positive tonal.

Time series is VTB (VTBR) shares price.

10.7717/peerjcs.1156/fig-B14 Figure B14 Left: most cumulative negative tonal words, Right: most common words with cumulative negative tonal.

Time series is VTB (VTBR) shares price.

10.7717/peerjcs.1156/table-C1 Table C1 Granger causality table.

Model: LDA, News: Kommersant.

SHARE	2015	2016	2017	2018	2019	2020	
LSRG	2.457	1.876	14.428 *	0.650	5.524 ***	2.953	
PLZL	5.293	22.273 *	0.316	1.761	3.704	9.180 **	
TATN	4.83	6.587	2.813	8.673 ***	16.706 *	17.682 *	
UPRO	NaN	NaN	9.346 **	8.710 ***	0.826	4.616	
CBOM	NaN	0.509	9.548 *	6.692	0.546	9.340 **	
FEES	10.803 **	3.578	8.978 **	1.104	4.114 ***	2.306 ***	
LKOH	9.241 **	8.753 *	13.236 *	3.427	10.876 *	9.581 **	
IRAO	0.051	12.546 *	4.097 *	12.986 *	2.419	9.230 ***	
AFLT	6.689	0.676	15.973 *	2.547	4.506	0.442	
HYDR	1.607	1.180	2.577	8.662 **	8.194 *	15.101 *	
AFKS	10.336 **	0.298	6.792 **	3.814 **	0.580	0.114	
GMKN	10.989 *	6.088	4.790 **	16.468 *	0.358	2.560	
SBERP	3.035	1.737	1.301	5.486	10.153 **	9.088 ***	
YNDX	1.385	0.396	1.219	9.283 **	2.122 ***	9.832 *	
FIVE	NaN	NaN	NaN	NaN	8.475 ***	2.390 ***	
POLY	17.176 *	8.288 ***	1.907	0.878	3.053	6.245	
SNGSP	17.13 *	12.821 *	13.244 *	8.096 **	2.777	1.090	
MGNT	8.870 ***	11.505 *	1.188	15.511 *	1.795	11.777 *	
NLMK	12.538 *	4.811	4.576	18.592 *	19.777 *	12.830 *	
DSKY	NaN	NaN	NaN	0.585	9.837 *	0.302	
ALRS	12.191 *	10.226 *	7.052 **	4.017	1.699	2.792 **	
MAGN	12.911 *	1.466	23.715 *	2.674	7.145 **	18.645 *	
TRNFP	12.059 *	5.525 ***	10.740 **	2.87	0.838	8.111	
MTSS	6.541	5.697 ***	7.765 ***	0.478	0.514	17.738 *	
SBER	1.662	4.164	0.462	4.178	5.920 **	1.296	
TATNP	6.013	2.785	8.950 *	4.905	7.666	12.254 *	
ROSN	0.254	9.393 **	4.195	4.697	11.711 *	20.003 *	
GAZP	0.145	11.537 *	5.956 ***	10.277 **	17.160 *	6.952	
PIKK	10.176 *	3.291 **	7.070	2.947 **	1.123	11.515 *	
QIWI	2.656	9.465 **	0.207	12.301 *	3.753	13.922 *	
MOEX	2.444	4.319 *	11.056 *	5.903 **	11.988 *	6.284	
RTKM	14.189 *	3.836	8.308 **	6.370 **	7.195 ***	13.747 *	
CHMF	4.874	6.981 ***	11.103 *	2.631	30.810 *	11.834 *	
RSTI	7.423	2.484	3.238	3.378	9.669 *	20.546 *	
RUAL	NaN	0.200	13.372 *	0.523	32.679 *	2.852	
SNGS	6.165	20.359 *	0.642	8.664 ***	3.602	5.866 **	
NVTK	12.123 *	12.167 *	8.508 *	5.281 **	5.693	3.858 *	
VTBR	13.448 *	2.458	4.823 **	5.014	7.234 ***	6.014	
PHOR	15.738 *	9.601 *	11.868 *	3.633	5.334 ***	4.197	
Notes.

* Significant at the p-value < 0.01.

** Significant at the p-value < 0.05.

*** Significant at the p-value < 0.10.

NaN lack of data on the issuer in the year under consideration

10.7717/peerjcs.1156/table-C2 Table C2 Granger causality table.

Model: LDA, News: Vedomosti.

SHARE	2015	2016	2017	2018	2019	2020	
LSRG	NaN	2.329 ***	18.43 *	5.199 **	19.236 *	18.591 *	
PLZL	NaN	12.415 *	11.956 *	3.358	1.862	0.914	
TATN	NaN	0.490	1.292	5.222	7.884 **	16.814 *	
UPRO	NaN	NaN	1.03	11.614 *	0.759	6.165	
CBOM	NaN	11.839 *	2.577 ***	0.974	7.505 *	9.780 **	
FEES	NaN	0.813	2.199 ***	10.264 *	7.039 ***	2.018	
LKOH	NaN	5.403 **	4.047 ***	1.045	7.104 ***	16.001 *	
IRAO	NaN	4.65	1.364	14.159 *	6.064 ***	16.530 *	
AFLT	NaN	4.305	3.632 **	15.368 *	5.929	17.160 *	
HYDR	NaN	3.553 **	3.987	17.756 *	2.122	11.92 *	
AFKS	NaN	3.247	13.130 *	11.185 *	22.509 *	9.728 **	
GMKN	NaN	3.379	7.706 *	13.253 *	14.021 *	7.904	
SBERP	NaN	10.352 **	0.936	8.467 ***	11.401 *	8.309 ***	
YNDX	NaN	5.894	9.761 *	0.302	8.797 ***	1.251	
FIVE	NaN	NaN	NaN	NaN	7.222 ***	4.781 **	
POLY	NaN	7.508	1.051	27.848 *	9.466 *	6.018	
SNGSP	NaN	6.849	4.262 ***	1.236	0.791	10.007 **	
MGNT	NaN	0.162	3.034 **	2.159	1.178	0.481	
NLMK	NaN	5.726 ***	5.778 *	6.157 ***	0.504	12.264 *	
DSKY	NaN	NaN	NaN	4.982	12.485 *	7.163	
ALRS	NaN	3.533	0.085	2.947 **	6.166	15.348 *	
MAGN	NaN	3.871	3.781	13.528 *	3.173	4.665 **	
TRNFP	NaN	2.808 **	1.543	1.635	8.003 **	16.716 *	
MTSS	NaN	3.619	1.839	14.828 *	11.205 *	11.636 *	
SBER	NaN	0.766	0.631	8.133 ***	6.288 *	8.302 ***	
TATNP	NaN	8.495 ***	4.398	0.385	1.579	17.92 *	
ROSN	NaN	2.815	3.862	1.207	1.543	9.274 **	
GAZP	NaN	4.556	3.974 ***	0.461	0.625	13.300 *	
PIKK	NaN	6.813	3.983 *	7.252 *	0.674	6.126	
QIWI	NaN	0.195	5.848	7.246	6.162 *	7.454 ***	
MOEX	NaN	0.427	0.626	5.644	4.730 **	1.881	
RTKM	NaN	14.967 *	7.092 **	8.240 *	3.389	10.968 **	
CHMF	NaN	2.65	11.483 *	4.646	3.976	21.677 *	
RSTI	NaN	2.414 ***	0.476	11.823 *	8.368 *	3.331	
RUAL	NaN	8.399 *	5.619 **	6.057 ***	6.557 **	2.875 **	
SNGS	NaN	28.047 *	7.732 *	6.530 **	13.924 *	7.120 *	
NVTK	NaN	4.307	8.313 *	1.281	0.392	13.484 *	
VTBR	NaN	2.260	2.643	1.621	5.589 **	9.195 ***	
PHOR	NaN	0.738	5.087	1.392	4.199 ***	0.294	
Notes.

* Significant at the p-value < 0.01.

** Significant at the p-value < 0.05.

*** Significant at the p-value < 0.10.

NaN lack of data on the issuer in the year under consideration

10.7717/peerjcs.1156/table-C3 Table C3 Granger causality table.

Model: LDA, News: RIA Novosti.

SHARE	2015	2016	2017	2018	2019	2020	
LSRG	7.157 *	4.324 *	3.121 **	0.299	1.575	6.876 ***	
PLZL	0.516	13.454 *	1.169	2.891	0.070	6.808 *	
TATN	8.955 **	6.167 *	0.453	6.466	9.375 **	9.318 **	
UPRO	NaN	NaN	14.449 *	1.474	2.218 ***	10.728 *	
CBOM	NaN	10.209 **	16.431 *	4.831	26.477 *	3.596	
FEES	10.076 **	7.653	1.803	0.349	4.289	0.206	
LKOH	15.407 *	9.489 *	15.926 *	2.857	3.184	8.986 **	
IRAO	2.480	4.467 *	8.873 ***	8.035	1.647	11.22 *	
AFLT	8.297 **	1.271	1.021	14.259 *	1.786	3.408 **	
HYDR	3.436	10.843 **	5.024 **	6.804	12.429 *	4.656	
AFKS	11.935 *	5.469 **	0.256	15.258 *	0.420	4.227 *	
GMKN	20.106 *	7.904 *	5.780	9.050 **	0.487	18.035 *	
SBERP	3.418	18.085 *	6.480	10.084 **	2.825 **	8.479 ***	
YNDX	13.352 *	2.266	2.030	7.604	5.124	3.497 **	
FIVE	NaN	NaN	NaN	NaN	13.566 *	1.015	
POLY	10.528 *	2.124 ***	27.809 *	1.022	3.455 **	3.049	
SNGSP	1.570	9.749 *	3.576	1.429	3.047	13.473 *	
MGNT	1.719	6.722 *	4.585 *	3.803	8.511 ***	1.222	
NLMK	10.761 **	10.172 **	5.064	0.668	0.517	7.802 **	
DSKY	NaN	NaN	NaN	14.059 *	5.612 ***	8.065	
ALRS	9.290 **	10.782 *	0.779	7.768	1.313	4.403	
MAGN	2.840	5.786	2.848	4.629	0.151	14.312 *	
TRNFP	4.892	2.899	3.990 *	8.989 ***	0.872	7.923 **	
MTSS	3.789	7.709 *	0.044	0.056	4.479 ***	0.871	
SBER	6.218	13.491 *	1.198	0.886	2.787 **	2.244 ***	
TATNP	6.430 *	4.630 **	7.549	3.912 ***	2.544	12.661 *	
ROSN	5.057	0.242	4.765	0.171	8.321 **	11.986 *	
GAZP	2.792	1.084	3.743 **	1.382	15.201 *	5.550	
PIKK	12.872 *	1.527	9.598 *	9.582 **	0.660	16.276 *	
QIWI	9.790 **	7.701 *	4.728 ***	4.209	0.513	6.117 *	
MOEX	10.241 **	4.772 **	3.578 **	8.028	6.706 *	6.299	
RTKM	8.398 **	6.542 *	5.676	1.285	0.878	6.882	
CHMF	12.138 *	4.899	0.520	1.194	1.158	38.428 *	
RSTI	1.275	11.289 *	2.424	2.624 ***	0.523	6.064	
RUAL	NaN	15.361 *	16.395 *	7.327	1.083	9.409 **	
SNGS	1.770	6.398 *	0.074	4.828	4.363	5.939	
NVTK	5.500 *	6.981 *	2.934	3.060	8.713 ***	1.819	
VTBR	10.067 *	0.625	2.059	4.229	8.844 ***	2.346 ***	
PHOR	11.960 *	8.211 *	3.403	0.160	10.680 *	12.231 *	
Notes.

* Significant at the p-value < 0.01.

** Significant at the p-value < 0.05.

*** Significant at the p-value < 0.10.

NaN lack of data on the issuer in the year under consideration

10.7717/peerjcs.1156/table-C4 Table C4 Granger causality table.

Model: DTM, News: Kommersant.

SHARE	2015	2016	2017	2018	2019	2020	
LSRG	1.229	2.267 ***	14.535 *	1.652	5.802 ***	2.807	
PLZL	5.559 ***	22.333 *	0.302	1.677	4.052	6.846 ***	
TATN	4.864	6.605	2.894	10.403 **	10.507 *	0.936	
UPRO	NaN	NaN	4.344	8.418 ***	1.373	4.324	
CBOM	NaN	2.963 **	5.030	7.992	1.937	9.221 ***	
FEES	10.662 **	3.501	8.853 **	2.083	2.867	1.520	
LKOH	12.859 *	12.341 *	14.087 *	5.681	4.215	1.093	
IRAO	0.065	13.307 *	4.035 *	13.340 *	3.537	0.979	
AFLT	6.948 ***	0.676	13.320 *	3.050	0.375	0.420	
HYDR	1.491	1.296	2.878 **	11.398 *	9.685 *	7.431	
AFKS	11.472 *	0.337	6.928 **	1.617	0.681	2.149 ***	
GMKN	7.287 ***	6.375	4.843 **	7.681 *	1.146	0.783	
SBERP	4.433	1.501	1.281	1.162	2.374	0.707	
YNDX	5.074	0.416	1.052	5.032	2.734	6.978 **	
FIVE	NaN	NaN	NaN	NaN	10.777 **	3.924 *	
POLY	15.356 *	8.005	1.865	0.868	1.939	5.670	
SNGSP	6.379 **	12.899 *	13.291 *	8.772 **	2.961	0.939	
MGNT	8.138 ***	7.366 ***	4.899 **	3.373	2.771	3.116	
NLMK	11.781 *	5.416	4.790	14.094 *	11.777 *	10.504 *	
DSKY	NaN	NaN	NaN	4.083 *	5.586 **	4.344 *	
ALRS	5.866 ***	11.824 *	6.812 **	4.252	1.773	2.652 ***	
MAGN	15.642 *	1.521	23.360 *	3.021	13.174 *	16.508 *	
TRNFP	10.064 *	5.551 ***	10.952 **	2.890	0.882	7.936	
MTSS	6.701	2.732 **	7.638 ***	0.561	0.57	12.877 *	
SBER	1.289	1.314	0.407	5.019	4.433 ***	5.005 *	
TATNP	7.066	2.719	9.140 *	5.022	1.508	2.331 ***	
ROSN	0.198	8.872 **	4.458	4.688	12.400 *	3.449	
GAZP	0.180	1.202	15.320 *	3.934 ***	9.076 **	5.304	
PIKK	10.548 *	3.221 **	7.092	2.640 ***	0.059	2.481	
QIWI	2.555	7.433 **	0.190	11.477 *	4.983	15.942 *	
MOEX	2.162	2.102 ***	11.570 *	11.276 *	8.733 **	1.709	
RTKM	16.927 *	4.159	7.140 ***	1.708	1.926	4.890	
CHMF	3.121	6.535	11.110 *	2.590	33.466 *	18.142 *	
RSTI	7.133	2.554	3.849 ***	1.329	17.397 *	7.280 **	
RUAL	NaN	0.046	13.752 *	0.744	32.069 *	2.788	
SNGS	5.399	20.976 *	0.678	9.024 ***	1.871	5.508 **	
NVTK	10.637 **	12.223 *	8.344 *	5.108 **	7.159	4.513 ***	
VTBR	12.734 *	2.534	6.443 **	5.166	7.574 ***	6.004	
PHOR	8.037 **	8.976 **	11.948 *	3.647	5.482 ***	4.455	
Notes.

* Significant at the p-value < 0.01.

** Significant at the p-value < 0.05.

*** Significant at the p-value < 0.10.

NaN lack of data on the issuer in the year under consideration

10.7717/peerjcs.1156/table-C5 Table C5 Granger causality table.

Model: DTM, News: Vedomosti.

SHARE	2015	2016	2017	2018	2019	2020	
LSRG	NaN	10.47 *	18.237 *	5.394 **	19.876 *	18.621 *	
PLZL	NaN	12.879 *	14.92 *	3.329	9.128 ***	0.951	
TATN	NaN	0.346	1.108	2.322	8.208 **	16.96 *	
UPRO	NaN	NaN	1.88	11.855 *	2.521	4.747	
CBOM	NaN	3.253	2.671 ***	2.148	7.848 *	9.880 **	
FEES	NaN	0.379	2.321 ***	10.540 *	7.541 **	11.927 *	
LKOH	NaN	4.590 ***	4.445 ***	1.098	7.562 ***	16.217 *	
IRAO	NaN	4.817	1.332	10.373 **	4.661	17.347 *	
AFLT	NaN	12.887 *	3.629 **	13.203 *	5.065	16.932 *	
HYDR	NaN	1.132	3.877	14.950 *	5.568	11.920 *	
AFKS	NaN	3.352	12.320 *	12.413 *	22.541 *	9.606 **	
GMKN	NaN	2.974	6.643	7.263 **	14.406 *	7.897	
SBERP	NaN	23.155 *	0.883	8.680 ***	11.739 *	8.435 ***	
YNDX	NaN	4.976	10.008 *	0.1980	8.302 ***	1.259	
FIVE	NaN	NaN	NaN	NaN	7.278 ***	3.877	
POLY	NaN	4.124	0.980	29.502 *	10.267 *	6.066	
SNGSP	NaN	4.200	4.656 **	0.818	0.601	10.250 **	
MGNT	NaN	0.195	3.604 **	5.103	0.728	0.424	
NLMK	NaN	5.884	6.997 *	5.288	0.821	12.669 *	
DSKY	NaN	NaN	NaN	3.025	5.879	2.110 ***	
ALRS	NaN	4.632	0.126	2.680 ***	1.267	13.337 *	
MAGN	NaN	2.469	6.846	10.097 **	3.034	4.658 **	
TRNFP	NaN	3.400 **	2.224 ***	1.430	7.918 **	16.716 *	
MTSS	NaN	2.586	5.560 ***	18.680 *	7.432 **	13.432 *	
SBER	NaN	0.214	0.607	8.122 ***	5.388 *	8.287 ***	
TATNP	NaN	9.566 **	5.816	0.509	1.634	18.212 *	
ROSN	NaN	6.207	3.962	1.139	1.634	9.630 **	
GAZP	NaN	8.257 ***	5.537 ***	1.095	0.618	13.328 *	
PIKK	NaN	3.805	3.815 **	8.180 *	0.602	6.285	
QIWI	NaN	2.560	6.054	8.279 ***	5.147 **	7.561 ***	
MOEX	NaN	7.338	0.576	6.080	4.690 **	1.881	
RTKM	NaN	16.871 *	5.886 ***	14.180 *	5.998	9.471 **	
CHMF	NaN	4.014	13.240 *	0.607	0.164	22.188 *	
RSTI	NaN	3.200 **	0.423	11.336 *	8.356 *	3.439	
RUAL	NaN	10.298 *	5.377 **	3.095 **	6.928 **	4.728 *	
SNGS	NaN	26.645 *	0.996	18.650 *	11.817 *	19.378 *	
NVTK	NaN	3.188	10.966 *	1.620	0.456	13.606 *	
VTBR	NaN	2.053	8.868 *	6.205	5.845 **	9.215 ***	
PHOR	NaN	4.800	4.506 *	0.544	3.673	1.780	
Notes.

* Significant at the p-value < 0.01.

** Significant at the p-value < 0.05.

*** Significant at the p-value < 0.10.

NaN lack of data on the issuer in the year under consideration

10.7717/peerjcs.1156/table-C6 Table C6 Granger causality table.

Model: DTM, News: RIA Novosti.

SHARE	2015	2016	2017	2018	2019	2020	
LSRG	10.376 *	2.805 **	3.358 **	0.154	5.827	7.051 ***	
PLZL	0.507	13.401 *	0.950	0.271	6.023	9.285 *	
TATN	9.279 **	6.169 *	0.454	7.071 ***	2.272 ***	8.687 ***	
UPRO	NaN	NaN	15.207 *	1.419	2.222 ***	10.738 *	
CBOM	NaN	10.417 **	16.329 *	5.870 ***	24.633 *	3.113	
FEES	10.636 **	8.179 ***	1.856	8.858 ***	4.086 ***	0.205	
LKOH	15.912 *	9.449 *	16.019 *	2.142	7.971 **	8.141 **	
IRAO	1.572	4.726 *	9.099 ***	7.711	5.061	10.568 **	
AFLT	8.192 **	1.258	1.287	0.598	1.924	3.502 **	
HYDR	4.253	4.102	4.848 **	11.400 *	9.306 **	4.737	
AFKS	12.387 *	5.687 **	0.310	5.263	4.548	4.526 *	
GMKN	20.052 *	7.724 *	6.154	10.727 **	2.542	17.878 *	
SBERP	3.231	17.807 *	5.202	6.108	2.870 **	8.526 ***	
YNDX	2.723	2.305	1.786	8.443 ***	5.573	3.591 **	
FIVE	NaN	NaN	NaN	NaN	8.515 ***	1.583	
POLY	10.608 *	2.102 ***	27.392 *	1.190	2.857 **	3.095	
SNGSP	1.344	9.470 **	8.649 ***	1.550	3.152	13.067 *	
MGNT	1.690	6.797 *	4.914 *	3.883	8.969 ***	1.342	
NLMK	11.101 *	10.600 **	6.049	10.597 **	0.542	8.113 **	
DSKY	NaN	NaN	NaN	11.760 *	5.846	6.180	
ALRS	8.953 ***	10.773 *	0.699	6.917	2.738	2.146 ***	
MAGN	2.863	5.846	3.366	1.783	1.837	14.168 *	
TRNFP	4.892	2.916	4.131 *	10.608 **	0.830	8.580 **	
MTSS	3.588	7.740 *	0.024	0.045	4.442 ***	0.888	
SBER	9.000 ***	13.487 *	1.022	3.329	2.924 **	1.927	
TATNP	6.403 *	4.649 **	7.348	6.957	0.437	12.200 *	
ROSN	4.863	0.461	5.637	4.471	1.078	12.237 *	
GAZP	8.739 ***	2.348	4.017 *	1.482	15.374 *	5.060	
PIKK	13.074 *	1.533	9.605 *	9.683 **	6.472	20.051 *	
QIWI	9.790 **	5.753 *	5.314 **	3.050	0.306	7.788 *	
MOEX	4.889	4.820 **	3.585 **	7.983	7.631 *	0.697	
RTKM	5.684	6.443 *	5.953	6.846 ***	0.510	6.396	
CHMF	12.142 *	5.333	1.636	5.487	0.326	39.896 *	
RSTI	1.272	11.291 *	2.530	2.976	1.677	6.098	
RUAL	NaN	18.023 *	19.341 *	7.185	1.060	9.435 **	
SNGS	6.182	6.230 *	0.167	5.227	4.643	5.860	
NVTK	5.520 *	7.053 *	3.078	3.429	8.472 ***	1.134	
VTBR	7.378 **	0.639	1.969	4.216	8.627 ***	2.366 ***	
PHOR	8.689 *	8.215 *	3.578	0.138	11.355 *	12.141 *	
Notes.

* Significant at the p-value < 0.01.

** Significant at the p-value < 0.05.

*** Significant at the p-value < 0.10.

NaN lack of data on the issuer in the year under consideration

Additional Information and Declarations

Competing Interests

Author Contributions

Data Availability

1 We use the standard 95%-quantile as a default value for α-level.

2 In this work, the LDA and DTM algorithms were used.

3 We use standard level of significance 5%.

4 As default value for Cj we use 0.3

5 We use the natural language toolkit (NLTK) in Python for tokenization tasks: https://www.nltk.org

6 We use a Python wrapper for an morphological analyzer for Russian language produced by Yandex Mystem for lemmatization task: https://pypi.org/project/pymystem3/

7 List of stop words is available from item 70 on https://www.nltk.org/nltk_data/

8 We use Granger causality test from statsmodel python-package: https://www.statsmodels.org

9 We use Augmented Dickey-Fuller unit root test from statsmodel python-package: https://www.statsmodels.org

10 We use Sharpe-ratio calculation from empyrical python-package: https://github.com/quantopian/empyrical

11 We use annual return calculation from empyrical python-package: https://github.com/quantopian/empyrical

12 We use LDA realization from gensim python package: https://radimrehurek.com/gensim/

13 We use C-language DTM realization: https://github.com/blei-lab/dtm

14 We use Word2Vec and Doc2vec realization from gensim python-package: https://radimrehurek.com/gensim/

15 All models realization from sklearn python-package: https://scikit-learn.org/stable/

16 We use Python realization, which is available on https://software.clapper.org/munkres/

17 We use Python realization of algorithm, which is available on https://pypi.org/project/kneed/ (Satopaa et al., 2011)

Coauthors Aleksei Riabykh, Denis Surzhko and Maxim Konovalikhin are employed at VTB but have no competing interests as VTB cannot either benefit or be at disadvantage from the findings of this research: no aspect of VTB activity/efficiency is studied in the article, and none of the possible alternative outcomes of the research would be preferred by VTB over other possible outcomes.

Aleksei Riabykh conceived and designed the experiments, performed the experiments, analyzed the data, performed the computation work, prepared figures and/or tables, authored or reviewed drafts of the article, and approved the final draft.

Denis Surzhko conceived and designed the experiments, performed the experiments, analyzed the data, performed the computation work, authored or reviewed drafts of the article, and approved the final draft.

Maxim Konovalikhin conceived and designed the experiments, performed the experiments, analyzed the data, performed the computation work, authored or reviewed drafts of the article, and approved the final draft.

Sergei Koltcov analyzed the data, prepared figures and/or tables, authored or reviewed drafts of the article, and approved the final draft.

The following information was supplied regarding data availability:

The code and data are available at Zenodo: skoltsov. (2022). hse-scila/-STTM: STTM_code (STTM_code_data). Zenodo. https://doi.org/10.5281/zenodo.7163701.

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
