# Peer review of "STTM: an efficient approach to estimating news impact on stock movement direction"

_PeerJ Computer Science, doi:10.7717/peerj-cs.1156_

## Round 0.1 · original submission · Major Revisions

Dear authors,

Thank you for submitting your work to PeerJ Computer Science. After completing the evaluation of your manuscript the reviewers recommend its reconsideration following major revision. I invite you to resubmit your manuscript after addressing the comments made by the reviewers. In particular:
1. Proofread your article for typos and grammatical mistakes (as suggested by both reviewers).
2. Provide a more appealing motivation and explanation for your work (as suggested by reviewer 1).
3. Extend and enhance the presentation of background concepts (as suggested by both reviewers).
4. Compare the results with a benchmark (as suggested by reviewer 1).
5. Include examples to better illustrate the scope and limitations of the proposed methodology (as suggested by reviewer 2).
6. Provide a discussion that helps to explain the observed behavior (as suggested by reviewer 2).

I hope you can complete the recommendation changes in a revision of your article.

Best,

Ana Maguitman

Reviewer 1 ·

Basic reporting

The paper analyzes the impact of news on the direction of changes in stock price, exploring machine learning techniques and using data from Russian media sources. From the prediction of the price direction, the study identifies a simple trading strategy and analyzes risk-adjusted performance metrics.

- The article is well-written, and the English language is adequate. The structure and format of the paper help the reader. However, a revision of the text may improve the article (i.e. "The figure D illustrate").

- The theoretical background could be enhanced since the literature on forecasting the direction of stock price movement is abundant. Some discussion about market efficiency could make the text more appealing, linking the practical approach of the study with finance theory. The research problem implies a hypothesis that the market is not efficient (at least in a semi-strong context).

- Some references are incomplete, such as those in lines 470, 482. Therefore, a careful review of the metadata of the references is important.

Experimental design

- The method is adequately described. In addition, many methodological issues are addressed in the paper, enriching the discussion.

- The study discusses Granger causality and risk-adjusted performance metrics and provides results for different combinations of algorithms and data sources, making the paper more interesting.

- The trading strategy from the study is based on a simple long position on stocks that are more likely to have a price appreciation over the next week. However, this strategy could be explained better. For instance, is there a recomposition of the portfolio each week? If yes, how is it done? Does the positioning in the stock take into account closing prices? Are transaction costs computed?

- Since the Russian language is not usually addressed in papers, it might be interesting for readers to have a brief description of the similarities, differences, and challenges in using machine learning to analyze languages derived from the Cyrillic alphabet.

Validity of the findings

- The study reports results from a variety of stocks, media sources, and algorithms. It would be interesting to compare the results with a benchmark, for instance, a buy-and-hold strategy or a broad stock market index.

- Therefore, the results could be more informative and more compelling for the readers.

Reviewer 2 ·

Basic reporting

This paper presents a method to relate the evolution of stock quotes and relevant textual economic information, aiming to provide a tool to maximize the gains of traders. They use close to two hundred thousand news articles (from Russian sources) and 39 stock time series from 2013 to 2021. They use
Sharpe's ratio ( representing the additional amount of return of a risky asset over a risk-free one that an investor receives per unit of increase in risk) to compare their model with other 26, drawn from the literature, finding that their approach performs better than the existing alternatives. Furthermore, for more than 70% of portfolio stocks they find that the Granger correlations established by their method are high, indicating its ability to capture the causal relation between news and stock values.

The paper is well written (with only a few scattered typos and infelicities). The literature review seems adequate and the graphics both conceptual and statistical are correct and help the reader to understand the procedure and the results.

On the other hand, the paper is rather terse, with very short discussions of highly relevant concepts, ranging from assumptions to conclusions.

Experimental design

The methodology seems to be sound, both in the treatment of textual information as well as in the use of time series. But the presentation of the experiments carried out and of the results obtained is hastily presented. I think some examples would be useful to understand the scope (and limitations) of the methodology presented here. I would also like to see some discussion of what could explain the drastic drop in causal power of the method in 2018 that can be seen in Figure 8.

Validity of the findings

While I can see the potential of this approach, I think that more information is needed to provide a sound assessment. Figure 9 is highly relevant here and thus deserves further discussion both about its meaning and on how a trader could make a difference using this approach.

---

## Round 0.2 · accepted · Accept

Thank you for systematically addressing the reviewers' comments and congratulations on the acceptance of your manuscript.

Reviewer 2 ·

Basic reporting

The authors have now provided the key insights I requested in my previous review. A casual reader can now, I think, make sense of some important results that were glossed over.

Experimental design

No comment.

Validity of the findings

No comment.

Additional comments

I'm satisfied with the responses to my queries. I think the paper exhibits clearly, in this new version, the importance of its content.

---

## Author Rebuttal · Round 0.2

Dear Dr Maguitman,

Thank you for giving us a chance to revise and resubmit our manuscript and for providing very helpful reviews.

We have carefully worked with all the comments and made new calculations needed to introduce an additional baseline, as suggested by the reviewer (now reflected in Fig. 9, Table 2 and discussion). We are now happy to submit the revised version of our paper with all major changes highlighted in red. Our point-by-point responses to the reviewers' concerns can be found in the table below.

Principal researcher at Laboratory for Social & Cognitive Informatics
and Associate Professor at HSE Campus in St. Petersburg, Department of Mathematics
National Research University Higher School of Economics

Dr. Koltcov Sergei

| COMMENT | RESPONSE |
|---|---|
| EDITOR'S COMMENTS | |
| Thank you for submitting your work to PeerJ Computer Science. After completing the evaluation of your manuscript the reviewers recommend its reconsideration following major revision. I invite you to resubmit your manuscript after addressing the comments made by the reviewers. In particular: | Thanks |
| 1. Proofread your article for typos and grammatical mistakes (as suggested by both reviewers). | Done |
| 2. Provide a more appealing motivation and explanation for your work (as suggested by reviewer 1). | Done; see par. 1 of the introduction |
| 3. Extend and enhance the presentation of background concepts (as suggested by both reviewers) | Overall extension of concept description has been suggested by reviewer 2 only. Unfortunately, it is unclear from her / his comment which concepts are desirable. However, the concept of market inefficiency specifically suggested by reviewer 1 has been introduced (see the previous point). |

| | |
|---|---|
| 4. Compare the results with a benchmark (as suggested by reviewer 1). | We did have a benchmark, but we have introduced a second one (Fig. 9, Table 5) along with all necessary calculations. Now we have to baselines: MOEX Russia index & Equal Weight Index. |
| 5. Include examples to better illustrate the scope and limitations of the proposed methodology (as suggested by reviewer 2). | In accordance with the recommendations of the second reviewer, we did the following. The scope of our approach is clarified in the new version of introduction. The limitation of our approach is presented in the new section 'Limitations'. Examples of topics, and the distribution of words in economic topics are shown in Figures B9-B14 (Appendix B). |
| 6. Provide a discussion that helps to explain the observed behavior (as suggested by reviewer 2). | On atypical models' behavior in 2018, the discussion is provided in lines 425-429. |
| REVIEWER 1 | |
| The paper analyzes the impact of news on the direction of changes in stock price, exploring machine learning techniques and using data from Russian media sources. From the prediction of the price direction, the study identifies a simple trading strategy and analyzes risk-adjusted performance metrics. | N/A |
| - The article is well-written, and the English language is adequate. The structure and format of the paper help the reader. However, a revision of the text may improve the article (i.e. "The figure D illustrate"). | Done |
| - The theoretical background could be enhanced since the literature on forecasting the direction of stock price movement is abundant. Some discussion about market efficiency could make the text more appealing, linking the practical approach of the study with finance theory. The research problem implies a hypothesis that the market is not efficient (at least in a semi-strong context). | Discussion about market efficiency has been added in the first paragraph of Introduction. |

| | |
|---|---|
| - Some references are incomplete, such as those in lines 470, 482. Therefore, a careful review of the metadata of the references is important. | Corrected |
| **Experimental design**<br><br>- The method is adequately described. In addition, many methodological issues are addressed in the paper, enriching the discussion.<br><br>- The study discusses Granger causality and risk-adjusted performance metrics and provides results for different combinations of algorithms and data sources, making the paper more interesting. | Thanks |
| - The trading strategy from the study is based on a simple long position on stocks that are more likely to have a price appreciation over the next week. However, this strategy could be explained better. For instance, is there a recomposition of the portfolio each week? If yes, how is it done? Does the positioning in the stock take into account closing prices? Are transaction costs computed? | The trading strategy description, including that of the recomposition procedure, has been extended, see "Evaluation procedure" section (lines 362-382). Since transaction costs are not computed, this information has been added to the Limitations section. |
| - Since the Russian language is not usually addressed in papers, it might be interesting for readers to have a brief description of the similarities, differences, and challenges in using machine learning to analyze languages derived from the Cyrillic alphabet. | Cyrillic alphabet is not an issue here, but the complexity of Slavic languages' grammar is. Now addressed in the News Preprocessing Pipeline section (lines 263-268) |
| **Validity of the findings**<br><br>- The study reports results from a variety of stocks, media sources, and algorithms. It would be interesting to compare the results with a benchmark, for instance, a buy-and-hold strategy or a broad stock market index.<br><br>- Therefore, the results could be more informative and more compelling for the readers. | Broad stock market index was already used in our research under the title of MOEX Russia Index; we have clarified that these indices are the same. Buy-and-hold strategy has been now added (see section Weakly Trading Strategy Performance), Figure 9 has been updated, and Table 2 added. Performance of both benchmarks is significantly below that of any other model tested. |
| REVIEWER 2 | |

| | |
|---|---|
| **Basic reporting**<br><br>This paper presents a method to relate the evolution of stock quotes and relevant textual economic information, aiming to provide a tool to maximize the gains of traders. They use close to two hundred thousand news articles (from Russian sources) and 39 stock time series from 2013 to 2021. They use<br><br>Sharpe's ratio ( representing the additional amount of return of a risky asset over a risk-free one that an investor receives per unit of increase in risk) to compare their model with other 26, drawn from the literature, finding that their approach performs better than the existing alternatives. Furthermore, for more than 70% of portfolio stocks they find that the Granger correlations established by their method are high, indicating its ability to capture the causal relation between news and stock values. | n/a |
| The paper is well written (with only a few scattered typos and infelicities). The literature review seems adequate and the graphics both conceptual and statistical are correct and help the reader to understand the procedure and the results. | Corrected |
| On the other hand, the paper is rather terse, with very short discussions of highly relevant concepts, ranging from assumptions to conclusions. | This paper is already more 3600 words long, leaving little room for lengthier conceptualizing. Additionally, it is not clear which exact concepts would be desirable. Nevertheless, the discussion about the concept of market inefficiency is added in para 1 of Introduction, as suggested by reviewer 1. |
| **Experimental design**<br><br>The methodology seems to be sound, both in the treatment of textual information as well as in the use of time series. But the presentation of the experiments carried out and of the results obtained is hastily presented. I think some examples would be useful to understand the scope (and | Our model is based on the use of a number of information sources that are related specifically to economic news. The use of irrelevant news does not allow us to use the proposed model effectively. All the limitations of our model are outlined in the new section 'Limitations'. Examples of topics, and the distribution of words in economic topics are shown in Figures B9-B14 (Appendix B). We also add additional baseline. So now we have two baselines:  MOEX Russia index & Equal Weight Index. |

| | |
|---|---|
| limitations) of the methodology presented here. | |
| I would also like to see some discussion of what could explain the drastic drop in causal power of the method in 2018 that can be seen in Figure 8. | We now discuss the most plausible explanation for this effect in the Discussion section, subsection: Granger Causality Tests, lines 420-430 |
| **Validity of the findings**<br><br>While I can see the potential of this approach, I think that more information is needed to provide a sound assessment. Figure 9 is highly relevant here and thus deserves further discussion both about its meaning and on how a trader could make a difference using this approach. | The meaning of Fig. 9 has been described more clearly in page 16, and its caption has been made much clearer. In-depth discussion of Fig. 9 is represented by the entire sub-section ('Weekly Trading Strategy Performance' that comprises most of the Discussion section). Both Figure 9 and the Discussion clearly show that our approach outperforms the two baselines and all other models, in terms of a set of quality metrics used. |